# Influence of the Web Formation of a Basic Layer of Medical Textiles on Their Functionality

**DOI:** 10.3390/polym14112258

**Published:** 2022-05-31

**Authors:** Gracija Čepič, Dunja Šajn Gorjanc

**Affiliations:** Department of Textiles, Graphics Art and Design, Faculty for Natural Sciences and Engineering, University of Ljubljana, Snežniška 5, 1000 Ljubljana, Slovenia; graci.325@gmail.com

**Keywords:** medical textiles, multilayer nonwovens, web formation process, breaking stress, permeability properties

## Abstract

The aim of the present study was to determine the influence of the spunbond process and the meltblown process, as well as various combinations of the two processes, on the functional performance of layered nonwovens for medical purposes. In the present study, eight samples used in the medical field, mainly for medical masks, were analysed. The samples studied were laminated nonwovens produced by the spunbond and meltblown processes, and combinations of spunbond and meltblown processes. In order to determine the influence of the technological process used to produce a base layer of nonwoven fabrics on their functionality, measurements of tensile strength and extension, water vapour permeability, air permeability, porosity, and thermal conductivity were performed. In addition, the structural characteristics of selected samples were analysed, such as fibre diameter, thickness, mass, raw material composition, and surface openness. The aim of the present study was to find the optimal combination of spunbond and meltblown processes for medical textiles. Based on the research results, we can conclude that the five-layer composite in which three layers are made by spunbond (S) and two layers are made by meltblown (M) in combination as SSMMS from PP fibres has optimal air permeability, filtration of pollutants passing through a protective mask, water vapour permeability and thermal conductivity, and is optimal for use as a multilayer nonwoven fabric for medical masks. Multilayer SSMMS composites also have a lower weight, resulting in less energy and time required for recycling such textiles.

## 1. Introduction

Textile materials are made from natural or synthetic fibers. With the development of new fibers and production technologies for the production of linear and flat textiles, the use of textile materials has increased significantly in various fields. One of the most important applications of textile materials is the medical textile industry. This new field represents a combination of textile technology and medical science with functional benefits. Nowadays, due to the increasing ageing population and the risks associated with human activities such as traffic accidents, chemical injuries, diseases, sports activities, etc., the demand for textile medical aids is rapidly increasing. These parameters have led to the rapid development of the medical textiles market as new materials, techniques, and technologies to produce advanced textile materials and medical devices. In general, textile materials have many unique properties such as good mechanical properties, elastic properties, permeability properties, absorbency and comfort, etc. These properties make the materials suitable for medical devices. In some cases, different textile materials or a combination of several materials are required to improve the properties of the textile product according to the end use. Various technological processes for manufacturing and finishing textiles can significantly improve certain properties of textile materials. These include water and blood absorption, wound healing and healing, etc. By using modern technologies, we can further improve the properties of textiles for medical purposes [1]. Fabric masks (e.g., woven masks) made of textile fabrics do not fall into the same category as medical (surgical) masks made of multilayer nonwoven fabrics and are therefore not considered suitable for medical purposes. For a mask to perform its function, the fabrics from which it is made must meet two basic requirements. First, the mask must provide an effective barrier to the transmission of droplets (≥5 µm) that are excreted through the respiratory tract and therefore travel at a higher rate than during normal breathing, which is referred to as filtration efficiency. The efficiency of the mask also depends on the compliance of the user [2].

This leads to the second requirement: the components of the mask should not interfere with the normal breathing process and should maintain a state of thermophysiological comfort in the facial region [2].

Nonwovens for medical applications offer numerous advantages, both in terms of user requirements and material properties. They ensure the safety of patients and medical staff by providing high levels of infection control, sterility and efficiency. Shorter production cycles, greater flexibility and versatility, and lower production costs are some of the reasons for the popularity of nonwovens in medical applications. In recent years, drylaid, meltblowing, spunbonding, hydroentangling, airlaying and carding, thermal bonding, and needlepunching have become important technologies for the production of nonwovens for medical applications [1,2,3,4,5,6,7,8,9,10,11,12,13,14].

## 2. Theoretical Part

### 2.1. Requirements and Materials

Medical textiles are a very important field of technical textiles. Medical textiles can be made of fibres, filaments, yarns, flat textiles and combinations (multilayer textiles). They are intended for first aid or clinical care of wounds and/or treatment of diseases. Textiles are an ideal interface between humans and medical devices, as they allow the combination of different types of fibres and thus different properties of textiles, the production of planar (2-D) and spatial (3-D) structures. This allows the production of different medical textiles depending on the requirements of the environment, the production of specific complex structures from natural and synthetic fibres. In some cases, the biodegradability of the fibres is also of great importance.

Medical devices include a wider range of products, from simple dressings to very complex devices to support human life. They are used in diagnostics, disease prevention, health monitoring and treatment of diseases, but also to improve the quality of life of people with disabilities. Medical devices in/on the body do not act chemically, i.e., they do not contain drugs [1,2,6,7,8].

This is the purpose of medical textiles: to improve health and well-being (e.g., use of hospital bedding with antimicrobial activity, surgical gowns, etc.), to treat external injuries (e.g., gauze, dressings for haemostasis and wound healing), to prevent infections (e.g., masks), to make implants (e.g., surgical sutures, vascular splints, artificial arteries), to support the growth of new tissue (e.g., Supporting tissue for tissue growth), as a carrier of active substances (e.g., nicotine patches), to assist in the treatment of injuries or to prevent their occurrence (e.g., support for the neck, back), to maintain hygiene (e.g., sanitary napkins, tampons, etc.) [1,2,6,7,8].

Medical textiles usually have to meet the following requirements: Non-toxicity (toxic substances are inorganic substances—copper, arsenic, silver, etc. and organic substances—drugs, pesticides, methyl alcohol), Non-carcinogenicity (carcinogenic substances—asbestos, coal dust, etc.), Non-allergenicity (allergenic substances on textiles—azo dyes, plasticizers, biocides, latex, etc.), Antimicrobiality (microorganisms—viruses, bacteria, moulds, protists. Microorganisms cause stains on textiles, deterioration of mechanical properties, unpleasant odours, diseases and nosocomial infections. The growth of microorganisms is inhibited by sterilization and disinfection), biodegradability (biodegradable fibres absorbed by the human body—cotton, viscose fibres, collagen, etc.), biocompatibility (biocompatibility refers to the prolonged contact of the implant material with the human body and blood without damaging the tissues and blood and without causing an immune reaction—rejection) [6,7,8].

According to the definition of medical textiles, the main components for processing textiles for medical purposes are fibres, yarns, fabrics and various types of composites. Mostly, medical textiles are based on five types of materials. These include woven, knitted, laminated and nonwoven materials or fibres. Recently, various polymers are used for medical textiles due to their unique properties such as versatility, biocompatibility, bioabsorption and non-toxicity. These materials are considered as basic materials for the production of various types of fibres or earlier for the production of medical textiles. Cellulose, chitin and chitosan, proteins such as gelatin and collagen, alginic acids and hyaluronic acids are classified as natural fibre forming polymers. Polyethylene terephthalate, polyamide, polyacrylonitrile, polypropylene, polyethylene, polyurethane, polyvinyl chloride, polyvinyl alcohol, polytetrafluoroethylene, aramide, aliphatic polyesters, polyanhydrides and polyamino acids are the main polymers. The ideal textile material for the medical field must meet certain requirements to accelerate the healing process and reduce side effects. Biocompatibility, good resistance to alkalis, acids and microorganisms, good dimensional stability, elasticity without contamination or staining, absorption/repellency and good air permeability are the most important properties of the medical material. As the main component of medical textiles, polymeric materials have a great influence on the biodegradability, biocompatibility, absorbency, antibacterial properties and other functional properties of the finished medical textiles. The possibility of recycling nonwovens for medical masks is also very important. In particular, mechanical and chemical recycling is possible. Multilayer nonwovens used as inner layers have a lower mass per unit area, which in turn leads to lower energy consumption during recycling [1,6,7,8]. The high productivity and extremely slow biotic degradation of medical textiles (medical masks) cause them to spread in the environment and harm wastewater. Plastics that enter the aquatic environment and their residues can remain for months, hundreds or thousands of years. During this time, they are defragmented by mechanical and photochemical processes, resulting in the formation of microplastics (<5 mm) or nanoplastics (<1 μm) [15]. Very important is also the collection of medical waste, which is already very well regulated in healthcare facilities.

Medical mask materials can be mechanically recycled and have the potential for industrial recovery. Considering the considerable heterogeneity of commercially available mask materials, different strategies could be introduced and new issues require further investigation [16].

In the present research, the focus is on medical textiles made of multilayer fibres or nonwovens.

Nonwovens belong to the group of unconventional textiles. They are made directly from fibres or from continuous filaments. The term nonwoven is often used as a general description for textiles made by a process other than weaving and knitting, or even more broadly for textiles that are not conventional textiles, paper rods, or plastics.

Medical textiles are usually made of cotton, polyethylene, polypropylene and polyurethane fibres.

Cotton is divided into natural, organic, cellulose and seed fibres. It is used for clothing (uniforms, underwear, pyjamas, etc.), personal care (bedding, diapers, towels, etc.), accessories (hats, scarves, shoes, etc.), hygienic and medical purposes (tampons, pads, gauze, absorbent cotton, masks, etc.), industrial purposes, etc. Cotton is temperature resistant, acid sensitive, hydrophilic, relatively resistant to alkalis and microorganisms, etc. Nonwovens made of cotton fibres are intended for the production of disposable products (sheets, bandages, cloths, towels, etc.) for hospitals and other medical purposes [12,13,14].

Polyethylene is classified into man-made, organic and polyaddition products [12,13,14]. In terms of its chemical structure, it is the simplest and most useful polymer. It is obtained by polymerization of ethylene. It is mechanically and chemically resistant (even at low temperatures), flexible, hygienic and environmentally friendly, lightweight, hydrophobic, elastic, etc. It is used for baby diapers, bags, pharmaceuticals, cosmetics, etc. [12,13,14].

Polypropylene is divided into man-made, organic and polyaddition products. It is obtained by polymerization of propene. It is very hydrophobic, has the lowest density (it is the lightest material), resistant to alkalis, very resistant to solvents, etc. It is used for food packaging, medical devices (disposable syringes, infusion bottles), cables, etc. [12,13,14].

Polyurethane or thermoplastic polyurethane is divided into man-made, organic and polycondensation products. Due to its high-performance properties, it is well known in medicine for advanced medical and physician purposes. It has good tensile strength, resistant to acids, fats and microorganisms, etc. It is mainly used for surgical masks, diagnostic equipment, dental materials, compression stockings, medical instrument cables, etc. [17,18].

### 2.2. Medical Masks and Multilayer Textiles

Disposable surgical masks are widely used by medical professionals, scientists and society. Since the outbreak of the COVID-19 pandemic, demand for these masks has increased as people believe they can use them to protect themselves from viral infections. SMS (Spunbond-Meltbond-Spunbond) structures are used for disposable surgical masks to provide 98% protection from bacteria and hydrophobicity to the user. SMS provides the highest level of protection and is the most popular combination structure, consisting of 1–5 gm^−2^ meltblown microfibers (MB) that are microporous and breathable. Surgical masks consist of a very fine middle layer with extra-fine glass fibres or synthetic microfibers covered on both sides by an acrylic-bonded, parallel-laid or wet-laid nonwoven fabric. The weights of the middle layers range from 10 to 100 gm^−2^, while the thickness of the fibres ranges from <1 to ±10 µm. Each layer has a different specific function: the middle layer serves as a filter, the outermost layer provides hydrophobicity, and the innermost layer serves as an absorbent to collect the drops coming from the users [19].

A surgical mask can be woven, nonwoven, or knitted. However, the non-woven method is the most common and costs less. Surgical face masks have been made from 20 gm^−2^ spunbond polypropylene, while 25 gm^−2^ nonwoven polypropylene is required in the meltblown process. Polystyrene, polyethylene, polyester, and polycarbonate can also be used to make surgical face masks. The filtration efficiency of the surgical mask is influenced by several factors, such as the selection of fibres, the manufacturing method, the structure of the nonwoven fabric, and the cross-sectional area of the fibres. Surgical face masks can be divided into three categories: Low barrier masks, medium barrier masks, and high barrier masks. A medium barrier mask type has a bacterial filtration efficiency (BFE) of ≥98%, while a low barrier mask has a BFE ≥ 95%. [20].

In the continuous process of manufacturing single-layer printing nonwovens and after combining them, various types of multilayer textiles can be produced at a production speed of 20 to 300 m·min^−1^ [12,13,14].

The so-called multilayer or layered textiles, which are a combination of spunbond and meltblown processes, should be mentioned here. There can be several such combinations, e.g., SSS—three-layer composite, where each layer is made by spunbond process, SSMMS—five-layer composite, where three layers are made by spunbond process and two by meltblown process, S—one layer by spunbond process, MB—one layer by meltblown process [12,13,14].

For heavier nonwovens, the carded process is used to produce the base layer. The process is based on the use of a puller with a roller to produce a single felt layer. For the production of heavier nonwovens, a horizontal cross-layer is integrated into the production line [12,13,14].

When carding the nonwoven from the fibre flocks, which serves as a template for the carding machine, the flocks are branched into individual fibres. The product of the carding machine is one or two nonwovens, which are joined together on the conveyor belt of the carding machine to form a single-ply sheet [12,13,14].

### 2.3. Overview of Research in the Field of Medical Textiles

Fibres or nonwovens are common materials used in all healthcare facilities. Medical textiles offer many opportunities for textile manufacturers, a range of textile materials in the form of knitted fabrics, woven fabrics, fibres and composites.

For a specific purpose, these products must meet the requirements of the end user. For both internal applications such as surgical sutures and implants, and external applications such as gauze, bandages, dressings, surgical masks, tampons, diapers, face masks, etc., they must achieve the highest standard properties.

New applications for medical textiles are directly related to innovations in new textile fibres, new materials and advanced production methods and technologies. Medical textiles are used in an attempt to improve the comfort of patients and end users [14].

The development of medical textiles in the world is progressing every day. In recent years, we have witnessed revolutionary achievements in the field of medical textiles. A new multilayer knitted structure with a coating of nanofibres was developed for an esophageal prosthesis. For this purpose, a yarn made of polyglycolic acid was knitted in tubular form and the surface was coated with polycaprolactane fibres [18,19]. In addition, breathable woven surgical garments with antimicrobial and liquid repellent properties were developed by coating silver and fluorocarbon nanoparticles using the pad drying method. The coating parameters were optimized accordingly [18,19,20,21,22].

In addition, the drug-infused antimicrobial silk suture material was developed for use in the closure and healing of wounds to prevent surgical infections. The material was found to have satisfactory approximation properties [23].

Other research focused on viscose fibres modified to increase the attraction of nano-metal oxides, namely aluminium oxide, zinc, or titanium oxide, to exert antimicrobial activity against two types of bacteria [24].

Nonwovens mainly composed of man-made fibres (single component, bicomponent) were presented at Techtextil 2019. They are bonded by a thermal process and can be recycled, i.e., returned to the technological production process.

Highly absorbent nonwoven strips were presented, which contain a highly absorbent foam in the core, which enables absorption of a large part of the fluid and thus faster wound healing, and at the same time protects against drying of the wound and has good permeability properties and thermal insulation, which prevents supercooled healing [25].

## 3. Materials and Methods

The basic objective of the present study was to determine the influence of the spunbond process and the meltblown process, as well as various combinations of the two processes, on the functional performance of multilayer nonwovens for medical purposes. As part of the study, eight samples used in the medical field, mainly for medical masks, were analysed. The samples studied were multilayer nonwovens produced by spunbond and meltblown processes, as well as combinations of spunbond and meltblown processes. In order to determine the influence of the technological process of manufacturing a base layer of multilayer nonwovens on their functionality, measurements of tensile strength and elongation, water vapour permeability, air permeability, porosity and thermal conductivity were performed. As part of the work, the structural characteristics of selected samples, such as fibre diameter, thickness, mass, composition of the raw material and openness of the surface were analysed.

For the experimental part of the study, eight samples of multilayer nonwoven fabrics produced by the spunbond and meltblown processes, as well as various combinations of the two processes, were investigated.

During the study, the fibre diameter, thickness, mass of the multilayer nonwovens, composition of the raw material, fibre diameter on SEM (scanning electron microscope) images of the sample cross-section, breaking force and breaking elongation, water vapour permeability, air permeability, porosity, and thermal conductivity of the nonwovens used as medical textiles were analysed.

### 3.1. Presentation of the Materials Analysed

The samples studied are multilayer nonwovens produced by the spunbond and meltblown processes and by various combinations of these two processes for medical and hygienic purposes. They are intended for health maintenance.

The fibres of the samples consisted of polyethylene, polypropylene and polyurethane (described in more detail in Section 2.2). The base layer of the samples was produced by the extrusion process, a combination of spunbond and meltblown processes (more in Section 2.3).

### 3.2. Fibre Diameter

Fibre diameter was measured using a scanning electron microscope (SEM), JSM-6060 LV Jeol; Japan, at 350× (five samples), 2000× (two samples) and 1600× (one sample) magnification. The thinner and thicker fibres of the two samples, were viewed at 350× and 2000× magnification to measure their diameters. Seven measurements of the fibre diameter were selected from the images of the fibres, from which the average diameter and the standard deviation and coefficient of variation were calculated.

### 3.3. Thickness

The thickness of the specimens was measured between two parallel plates at a pressure of 20 cN/cm^2^ and according to the standard ISO 9073-2: 1995. Five thickness measurements in [mm] were made on each specimen [26].

### 3.4. Mass

The mass of a flat textile is expressed as mass per unit area in g/m^2^. The mass of the samples was measured according to the standard ISO 9073-1: 1989 [27]. The three measurements were performed on samples with an area of 550 cm^2^ (length—27.5 cm and width—20 cm).

### 3.5. Composition of the Raw Material

The composition of the raw material is given in Table 1.

Table 1 shows that the samples consist of polypropylene, polyethylene and polyurethane fibres. Most of them contain polypropylene fibres. Samples 3 to 6 contain polypropylene fibres. Samples 1 and 2 consist of a mixture of polypropylene and polyethylene fibres. Polyethylene contains sample 7, TPU or thermoplastic polyurethane contains sample 8.

### 3.6. Image of Materials Used on SEM Microscope

The image of the materials used on a scanning electron microscope (SEM), JSM-6060 LV Jeol; Japan, is shown in Table 4. There, the fibres with different diameters, the process of producing the base layer, and the shape of the fibre cross-section were considered in more detail.

### 3.7. Tensile Properties

Tensile properties were determined on a constant strain rate dynamometer; Instron 6022. A specimen 25 mm wide and 150 mm long was inserted into the upper and lower clamps with a clamp length of 100 mm. Elongation was performed at a rate of 100 mm/min with a measurement step of 1 mm. The tensile force and tensile strain of each material were measured on four specimens, two in the longitudinal direction and two in the width direction [28].

Tensile properties were measured according to ISO 9073-3: 1989. [25] The breaking stress was calculated according to Equation (1):(1)$$\sigma =\frac{F_{pr}}{A}=\frac{F_{pr}}{d\cdot w}\;\left[\mathrm{MPa}\right]$$
where are: *σ*—breaking stress [MPa]; *F_pr_*—breaking force [N]; *d*—thickness of the sample [mm]; *w*—width of the sample when measuring the breaking force [mm].

### 3.8. Water Vapour Permeability

The measurement of water vapour permeability was performed according to ASTM E96: E96M method [29] by placing a test specimen with an area equal to the opening of the measuring cup. It was covered it with a lid with an opening of 3 cm in diameter. The cup contained 7 mL of water. One hour after placing the sample, the mass of all elements together (mass of the water cup and the sample) was weighed. After weighing, the samples were left in the laboratory for 24 h. After 24 h, all elements were weighed again and the water vapour permeability was calculated from the difference of the masses. [30]. The water vapour permeability rate was calculated using the following Equation (2):(2)$$WVT=\frac{\Delta m}{S\cdot T}\;[g/m^{2}]$$
where are: *WVT*—water vapour permeability rate [g/m^2^h]; Δ*m*—difference between the weight of the cup with water and the sample after one h and after 24 h, [g]; *S*—surface area of the lid opening [m^2^]; *T*—time [h].

### 3.9. Air Permeability

Air permeability was measured according to the standard SIST EN ISO 9237: 1999 [30] by passing air through flat textiles attached to the device MESDAN LAB. The device was then used to measure of air drawn in that penetrates through the surface of the clamped material at a given pressure. The measurement was made on a test area of 10 cm^2^ and at a pressure of 100 Pa. Air permeability could be measured for all eight materials after 5 measurements [30].

Air permeability was calculated according to Equation (3):(3)$$Q=\frac{q}{6F\;}\;[m^{3}/m^{2}\min]$$
where are: *Q*—amount of air extracted in m^3^ in 1 min, calculated per 1 m^2^ at 20 mm water column [m^3^/m^2^ min]; *q*—volume of air flowing through the test body (value read on the instrument) [l/h]; *F*—test area [cm^2^].

### 3.10. Porosity

A body is porous if its interior contains a certain proportion of empty/airy spaces or pores in the total volume of the body. The air spaces may be interconnected, partially interconnected or completely unconnected. Depending on how the air spaces are interconnected, we speak of a permeable or impermeable porous body. Porosity is more precisely defined by a set of parameters that describe the internal geometric structure of a body [31,32,33,34].

The size and distribution of pores were determined by the Jakšič method. A rotameter is used to measure the volumetric velocity of air flow through a given area of the dry sample at different pressure differences [30].

The porosity meter consists of a compressor that generates a positive pressure of 8 bar, a reducing valve that reduces the pressure to 2.5 bar, and an auxiliary vessel to which a rotameter with a measuring range of up to 833 cm^3^/s and a manometer are connected. Between the auxiliary vessel and the rotameter is a valve for fine regulation of the pressure. The moving air flows through the rotameter into the measuring element to which the sample is attached. The measuring head allows the clamping of 1.5 and 10 cm samples.

To measure porosity, the samples were first prepared. For each sample, three dry and three wet (soaked in distilled water) samples were prepared. The sample was clamped in the 1.5 cm measuring head and the measurement was started.

### 3.11. Thermal Conductivity

Thermal conductivity was measured using a comparative method in accordance with DIN 52 612 (Part 1) [35]. To determine thermal conductivity, one test specimen per sample was cut to the dimensions of the measuring head of the instrument. The instrument has two blocks that are heated to a certain temperature so that the heat flow from the lower heating block flows upwards to the colder one. In the process, the heat flow passes through the reference sample (glass plate) and the fibre sample. The temperature differences are measured with thermocouples in three measuring copper plates, connected to the ALMEMO 2590 measuring instrument, from which the temperature values of the measuring copper plates are read.

The thermal conductivity is inversely proportional to the temperature differences and is calculated using Equation (4):(4)$$\lambda_{x}=\lambda_{n}\cdot \frac{d_{x}}{d_{n}}\cdot \frac{\left(T_{3}-T_{2}\right)}{\left(T_{2}-T_{1}\right)}\;\left[\frac{W}{\mathrm{mK}}\right]\;$$
where are: *λ_x_*—thermal conductivity of the test sample, [W/mK], *λ_n_*—thermal conductivity of the reference glass plate, [*λ_n_* = 1.0319 W/mK], *d_x_*—thickness of the test sample, [mm], *d_n_*—thickness of the reference glass plate, [*d_n_* = 4 mm], *T*_1_—temperature of the colder thick copper plate, [°C], *T*_2_—temperature of the middle thin copper plate, [°C], *T*_3_—temperature of the warmer thick copper sheet, [°C].

### 3.12. Statistical Analysis

The influence of the technology of the nonwoven forming process (spunbond, meltblown and combinations of both processes) on the functional properties of the studied fabrics (breaking strength and elongation, water vapor permeability, air permeability, thermal conductivity, porosity) was investigated using analysis of variance (ANOVA) to determine the significance of the technology of the nonwoven forming process on the functional properties of the multilayer samples, which differ according to the process used to produce the base layer (spunbond, meltblown and their combinations).

The base of a single factorial ANOVA is represented by dividing the sums of squares into between classes (SSb) and within classes (SSw). This technique allows all classes to be compared simultaneously rather than considering them individually. This method also assumes that the samples are normally distributed. The single factorial analysis is calculated in three steps. The sums of squares are first determined for all samples and then for the within-class and between-class cases. Degrees of freedom (df) are also determined of each stage, where df is the number of independent pieces of information involved in estimating a parameter. These calculations are used in conjunction with Fisher statistic to analyze the null hypothesis. The null hypothesis states that there are no differences between the means of the different classes, which means that the variance of the within-class samples should be identical to the variance of the across-class samples. If F ≥ 1, then it is likely that there are differences between the class means. These results are then tested for statistical significance or the *p*-value, where the *p*-value is the probability that a variate takes on a value that is greater than or equal to the value observed purely by chance. If the *p*-value is low (e.g., *p* ≤ 0.05 or *p* ≤ 5%), the null hypothesis is rejected, indicating that differences exist between classes and that these differences are statistically significant. If the *p*-value is greater than 0.05 (e.g., *p* ≥ 0.05 or *p* ≥ 5%), the null hypothesis is accepted, meaning that the differences between classes are random. ANOVA was created using SPSS Statistical software [36].

## 4. Results

### 4.1. Fibre Diameter

The results of the fibre diameter values are presented in Table 2.

### 4.2. Thickness and Mass

The thickness and mass results are presented in Table 3.

### 4.3. Display of Used Materials on SEM Microscope

The Table 4 shows the microscopic appearance of the samples at two different magnifications.

### 4.4. Breaking Stress and Extension

The results of the breaking stress and extension in the longitudinal and transverse directions are shown in Figure 1, Figure 2 and Figure 3.

### 4.5. Water Vapour Permeability and Air Permeability

The results of water vapour permeability and air permeability are shown in Figure 4.

### 4.6. Porosity

The results of porosity are listed in Table 5.

### 4.7. Thermal Conductivity

The results of thermal conductivity are shown in Figure 5.

### 4.8. Statistical Analysis

The single factor ANOVA is used to determine the significance of the technology of web formation process (spunbond, SSMMS, meltblown) on the functional properties of multilayered samples for medical textiles (*p* value < 0.05) (Table 6).

## 5. Discussion

### 5.1. The Discussion of Fibre Diameter Results

The highest fibre diameter was measured in sample 2 with an average of 19.57 μm and the lowest in sample 6 with thinner fibres, namely 1.86 μm. Regarding the fibre diameter, the results also differ in the production of the base layer of the nonwovens. In the spunbond process, the fibre diameter ranges from 11.53 μm (sample 3) to 19.57 μm (sample 2). Thinner and thicker fibres are visible in the five-layer composite, which is a combination of spunbond and meltblown processes. The diameter of the thinner fibres ranges from 1.86 μm (sample 6 A) to 2.40 μm (sample 5 A), and the diameter of the thicker fibres ranges from 12.84 μm (sample 6 B) to 17.414 μm (sample 5 B). In the meltblown process, the fibre diameter ranges from 2.81 μm (sample 7) to 3.55 μm (sample 8). Sample 2 has the highest fibre diameter with an average of 19.57 μm and sample 7 has the lowest with 2.81 μm.

### 5.2. The Discussion of Thickness and Mass Results

From the results of the thickness measurement of the samples, it can be concluded that the samples have different thicknesses, ranging from 0.063 mm (sample 6) to 0.366 mm (sample 1). For the samples that were produced by the spunbond process, the thickness ranges from 0.116 mm (sample 3) to 0.366 mm (sample 1). The thickness of the five-layer composites—SSMMS produced by a combination of spunbond and meltblown processes ranges from 0.063 mm (Sample 6) to 0.142 mm (Sample 5). For the samples produced by the meltblown process, the thickness is quite similar, namely sample 7: 0.186 mm and sample 8: 0.172 mm.

From the results of the mass of the samples, it can be concluded that sample 8 (46.73 g/m^2^) has the highest mass in the meltblown process, and sample 6 (7.64 g/m^2^) has the lowest mass in the five-layer spunbond and meltblown processes. In the spunbond process, sample 1 (30.91 g/m^2^) has the highest mass, in the five-layer spunbond and meltblown process, sample 5 (15.27 g/m^2^) has the highest mass, and in the meltblown process, sample 8 (46.73 g/m^2^) has the highest mass. In the spunbond process, sample 3 (13.27 g/m^2^), in the five-layer spunbond and meltblown process sample 6 (7.64 g/m^2^) and in the meltblown process sample 7 (27.82 g/m^2^) have the lowest surface mass.

### 5.3. The Discussion of Breaking Stress and Extension Results

The highest breaking stress in the longitudinal direction occurs in sample 3 (2.279 MPa), which was produced by the triple spunbond process and the lowest in sample 1 (0.858 MPa) which was produced by the single-layer spunbond process (Figure 1). Sample 3 has the lowest thickness (0.116 mm) and fibre diameter (11.53 µm), while sample 1 has the highest thickness (0.366 mm) and very high fibre diameter (19.34 µm). Sample 3 consists of three spunbond layers, while sample 1 consists of only one spunbond layer.

For the samples produced by the five-layer spunbond and meltblown process, the highest breaking stress is calculated for sample 6 (1.422 MPa) and the lowest for sample 5 (1.332 MPa). Sample 6 has a lower thickness (0.063 mm) and fibre diameter (A: 1.86 µm; B: 12.84 µm) than sample 5 (thickness = 0.142 mm; fibre diameter = A: 2.40 µm; B: 17.41 µm).

For the meltblown process, the highest breaking stress is calculated for sample 8 (1.865 MPa) and the lowest for sample 7 (0.262 MPa). Sample 8 has a smaller thickness (0.172 mm) than sample 7 (0.186 mm) but a larger fibre diameter (3.55 µm), while sample 7 has a fibre diameter of 2.814 µm.

From the results of the measurement of tensile force, breaking stress and extension in the transverse direction, we can conclude that among the spunbond samples, the highest tensile stress occurs in sample 3 (4.514 MPa) and the lowest in sample 1 (1.634 MPa). Sample 3 has the lowest thickness (0.116 mm) but consists of three spunbond layers, while sample 1 has the highest thickness (0.366 mm) of all analysed samples but consists of only one spunbond layer.

For the five-layer spunbond and meltblown process, the highest breaking stress is measured for sample 6 (5.860 MPa) and the lowest for sample 5 (3.921 MPa). Sample 6 has a lower thickness (0.063 mm) than sample 5 (0.142 mm).

In the meltblown process, sample 8 has the highest breaking stress (2.290 MPa) and sample 7 (0.273 MPa) has the lowest. Sample 8 has a smaller thickness (0.172 mm) than sample 7 (0.186 mm) but a larger fibre diameter (3.55 µm), while sample 7 has a fibre diameter of 2.814 µm.

From the results, it can be seen that for all the samples, both in longitudinal and transverse direction, the thickness affects the breaking stress, as the sample with the highest breaking stress has a lower thickness than the sample with the lowest breaking stress, which has a higher thickness. For the spunbonded samples, the number of layers also affects the higher breaking stress, and for the meltblown samples, the fibre diameter also affects the breaking stress. The samples produced by the SSMMS multilayer process have a higher breaking stress than the other samples because they consist of five layers.

Statistical analysis single factor ANOVA shows the statistically significant influence of the technology of the web formation process on the breaking stress and extension (*p* value < 0.05) (see Table 6).

### 5.4. The Discussion of Water Vapour Permeability Results

From the obtained results, it can be concluded that the highest water vapour permeability is preset in sample 1 (83.43 g/m^2^h), whose surface mass is 30.91 g/m^2^ (Figure 2). Sample 1 consists of a spunbond layer, whose structure affects higher water vapor permeability, and has a large fibre diameter (19.34 μm). Sample 7 has the lowest water vapour permeability (71.64 g/m^2^h), whose surface mass is 27.82 g/m^2^. Sample 7 consists of a single meltblown layer and has a fibre diameter of 2.81 μm.

Although the samples are produced by different processes, they have very different fibre diameters and thicknesses, which does not significantly affect the water vapour permeability itself. The only major difference is between the highest and lowest vapour permeability, as sample 1 was made by the spunbond process and sample 7 by the meltblown process. On average, the spunbonded samples have higher water vapour permeability than the samples made by the meltblown process. The spunbond samples have a larger fibre diameter and consequently a smaller specific surface area, resulting in a higher water vapour permeability than the meltblown samples, which have a much lower fibre diameter. The values of the water vapour permeability of the samples produced by the five-layer SSMMS process are between those of the samples produced by the spunbond process and those produced by the meltblown process.

Sample 1 (38.84 m^3^/m^2^min) has the highest air permeability, followed by samples 2, 3, 4, 6 and 5. Sample 7 (4.97 m^3^/m^2^min) and sample 8 (2.28 m^3^/m^2^min) are the least permeable. The results show that the spunbond samples have higher air permeability, the five-layer spunbond and meltblown composites are in the middle and the meltblown samples have the lowest air permeability. It can be seen that the nonwoven formation process, fibre diameter and thickness affect the air permeability.

The spunbond samples have a larger fibre diameter and consequently a smaller specific surface area, resulting in higher air permeability than the meltblown samples, which have a much smaller fibre diameter.

The statistical analysis single factor ANOVA shows the statistically significant influence of the technology of web formation process on the air permeability (*p* value < 0.05) (see Table 6). On the other hand, the statistical analysis shows that the influence of the technology of web formation process on the water vapour permeability-WVT is not significant (*p* value > 0.05) (see Table 6).

### 5.5. The Discussion of Porosity Results

The average pore diameter ranges from 23.99 μm to 198.59 μm, the surface openness from 3.17% to 70.77%, and the maximum pore diameter of the bubble point from 90 μm to 1049 μm (Table 5).

Sample 7 (70.77%, 1049 μm) has the highest bubble point and surface openness, and sample 8 (3.17%, 90 μm) has the lowest. The openness of the surface depends on the base layer web formation process, the web bonding process, the length of the fibres, the diameter of the fibres, and the orientation of the fibres in the web. Samples 7 and 8, which had the highest (sample 7) and lowest (sample 8) surface openness, respectively, were both produced by the meltblown process. The results were surprising, as samples 7 and 8 have the lowest air permeability (Figure 3). We suspect that sample 7 has the highest surface resistivity because it has the smallest fibre diameter (2.814 μm) and most likely consists of slightly shorter fibres, which we suspect is the case based on the SEM image of the sample in Table 4. This is most likely the reason for the highest mean open surface area (70.76%).

The statistical analysis shows non-significant influence of the of the technology of web formation process on the porosity—openness of surface (*p* value > 0.05) (see Table 6).

### 5.6. The Discussion of Thermal Conductivity Results

Sample 6 has the highest thermal conductivity (0.311 W/mK), which means that it provides the lowest thermal insulation, as the sample has the lowest thickness (0.063 mm) and surface mass (7.64 g/m^2^) (Figure 3). It can be noted that the thermal conductivity of the other studied samples ranges from 0.041 W/mK to 0.064 W/mK, which means that the combination of spunbond and meltblown processes does not significantly affect the thermal conductivity. The thickness and surface mass of the samples are lower for sample 6, which thus has the highest thermal conductivity. Sample 3 with the lowest thermal conductivity of 0.041 W/mK provides the highest thermal insulation.

The statistical analysis shows a non-significant influence of the technology of the web formation process on the thermal conductivity (*p* value > 0.05) (see Table 6).

## 6. Conclusions

In the present work, we have investigated the influence of spunbond and meltblown processes and their combinations on the functionality of multilayer nonwovens for medical purposes.

From the results, we can conclude that the number of layers affects the value of breaking stress for the samples prepared by spunbond process. The samples produced by the spunbond process have a higher average breaking stress than the samples produced by the SSMMS and meltblown processes.

The reason for the lower breaking stress of the investigated samples is due to the nonwoven structure, which is thermally bonded and made directly from fibres. The studied samples have a low surface mass and thickness and consequently a lower value for the breaking stress.

The spunbonded samples have, on average, higher water vapour permeability than the meltblown samples. The spunbonded samples have a higher fibre diameter and consequently a lower specific surface area, which affects the higher water vapour permeability than the meltblown samples, which have a much smaller fibre diameter.

The samples have different air permeability. The samples produced by the spunbond process have the highest values. The samples produced by the meltblown process have the lowest values. We can conclude that the reason for these different results is only the process used to produce the nonwoven fabric. The samples produced by the spunbond process have a higher fibre diameter and consequently a lower specific surface area, resulting in higher air permeability than the samples produced by the meltblown process, which have a much lower fibre diameter.

Based on the results for porosity, we found that the openness of the surface depends on several factors, such as the base layer web formation process, the fibre diameter, the length of the fibres, and the orientation of the fibres in the web. In our study, the base layer manufacturing process or the combination of spunbond and meltblown processes does not significantly affect the porosity, but depends on other factors. We assume that the samples with the largest surface openings have the smallest fibre diameter (about 2.814 μm) and in all likelihood are composed of slightly shorter fibres, which we suspect from the sample image in Table 4. This is most likely the reason for the highest mean open area of this type of samples (70.77%).

From the thermal conductivity results, we can conclude that all samples have low thermal conductivity except for sample 6, which has much higher thermal conductivity compared to the other samples because it has the lowest thickness (0.063 mm) and surface mass (7.64 g/m^2^). The combinations of spunbond and meltblown processes do not affect the thermal conductivity, while the thickness and surface mass of the samples are more affected.

From the experimental results, it can be concluded that three-layer spunbond samples—SSS samples and five-layered composite spunbond and meltblown samples—SSMMS samples have the optimum properties for use as medical textiles. Three-layer spunbond samples—SSS samples are made of polypropylene fibres and consist of three spunbond layers, while SSMMS samples consist of three spunbond layers and two meltblown layers. The multilayer samples have average thickness (0.11 mm to 0.21 mm), fibre diameter (11 μm to 17 μm; thinner fibres 2.4 μm), surface mass (13 g/m^2^ to 25.7 g/m^2^), the values of breaking stress of the samples range from 2.3 MPa to 4.5 MPa, the openness of the surface area of the samples ranges from 30% to 47%, and the thermal conductivity ranges from 0.04 W/mK to 0.05 W/mK.

Statistical analysis shows that the technology of the web formation process has only a statistically significant effect on the breaking stress, extension and air permeability.

The requirements for such textiles are low thickness, high tensile strength, optimal water vapour permeability, air permeability, porosity, and good thermal conductivity to provide maximum comfort to the user.

Important findings of the research are in the direction of an optimal combination of spunbond and meltblown processes of multilayer composite for medical masks with optimal permeability and lower surface mass, which reduces energy consumption when recycling medical masks from such textiles.

Collecting and recycling medical masks is very important in the future to reduce the amount of this waste in the environment. Masks are made of polymers that can be easily recycled and it is important to establish a proper recycling protocol. On the other hand, it is also important to reduce the mass of this waste, which is achieved by a combination of spunbond and meltblown processes and improves the useful properties of medical masks.

## Figures and Tables

**Figure 1 polymers-14-02258-f001:**
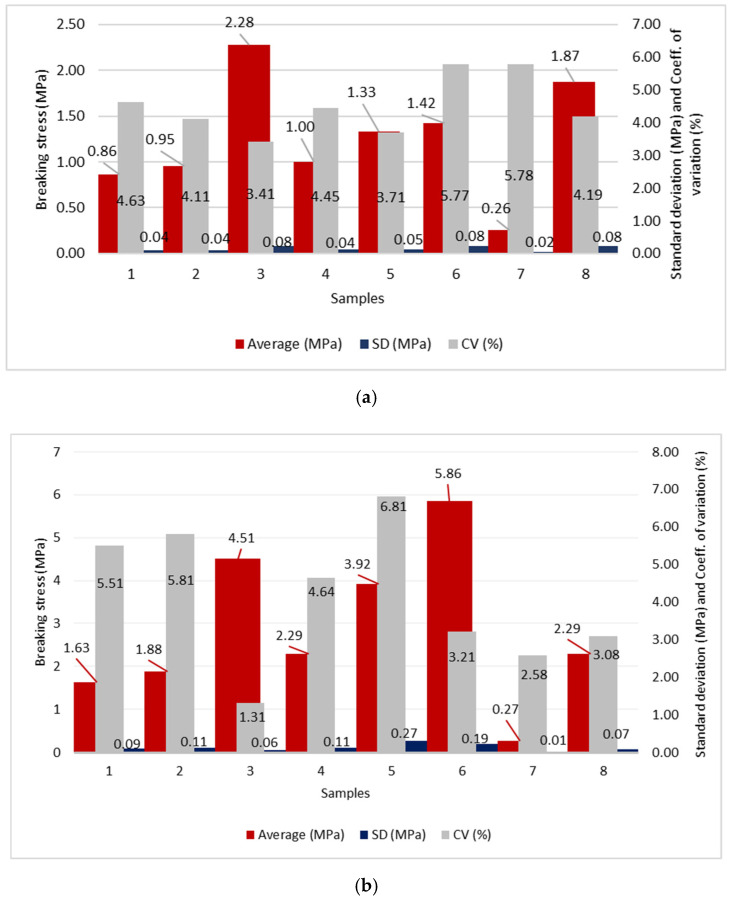
Breaking stress; (**a**) in longitudinal direction and (**b**) in transverse direction; Average breaking stress in MPa; SD—standard deviation in MPa; CV—coefficient of variation in %.

**Figure 2 polymers-14-02258-f002:**
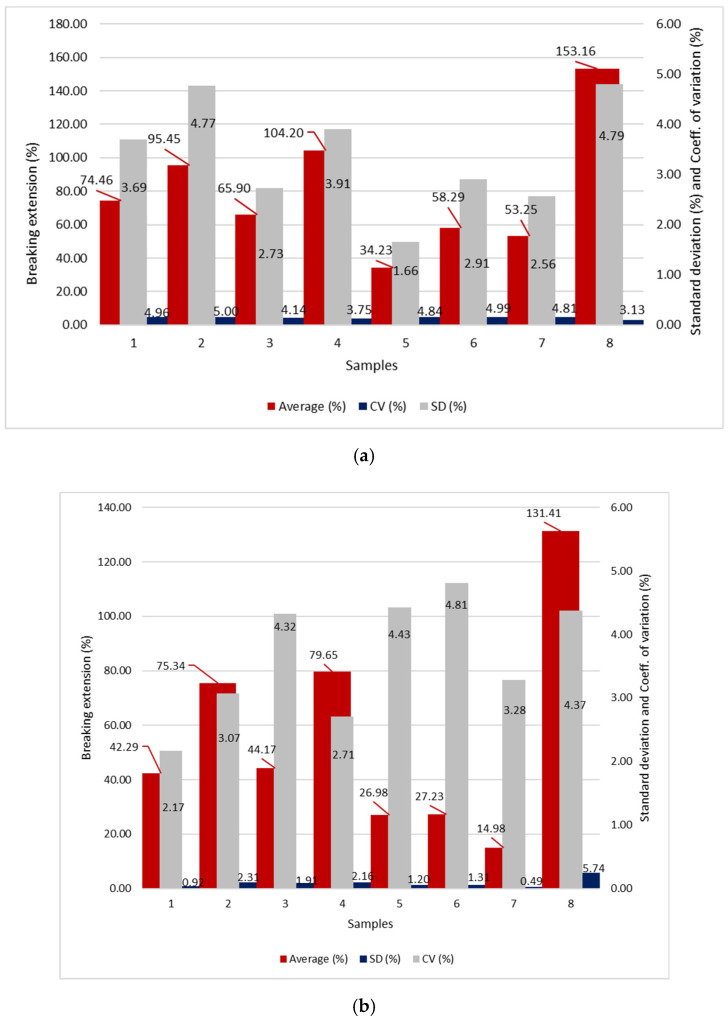
Breaking extension; (**a**) in longitudinal direction and (**b**) in transverse direction; Average breaking extension in %; SD—standard deviation in %; CV—coefficient of variation in %.

**Figure 3 polymers-14-02258-f003:**
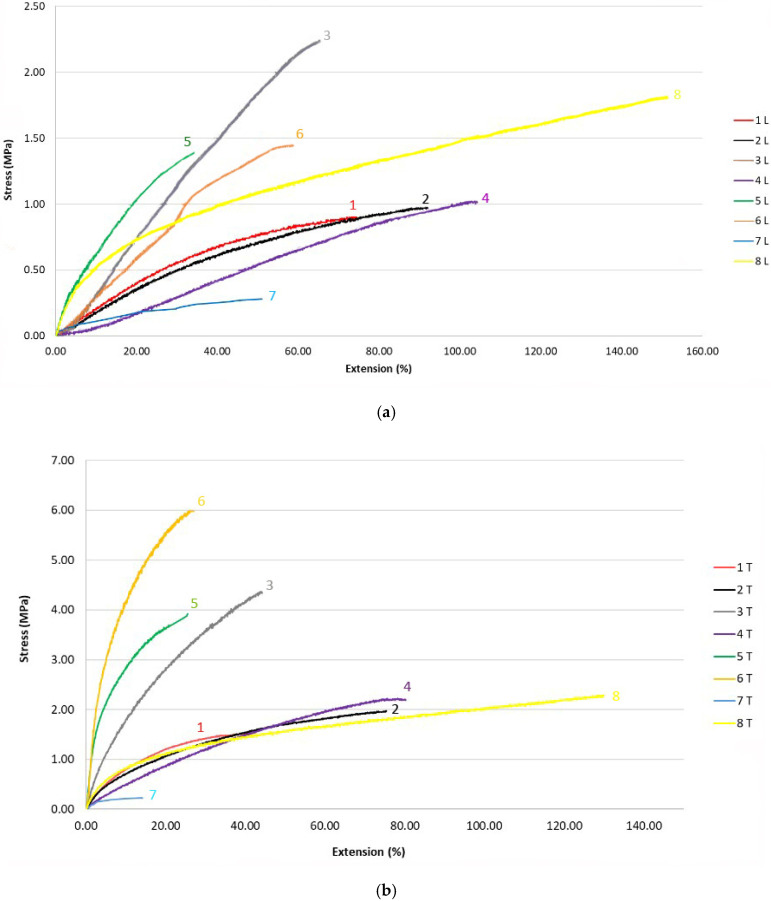
Stress-extension curves; (**a**) in longitudinal direction and (**b**) in transverse direction.

**Figure 4 polymers-14-02258-f004:**
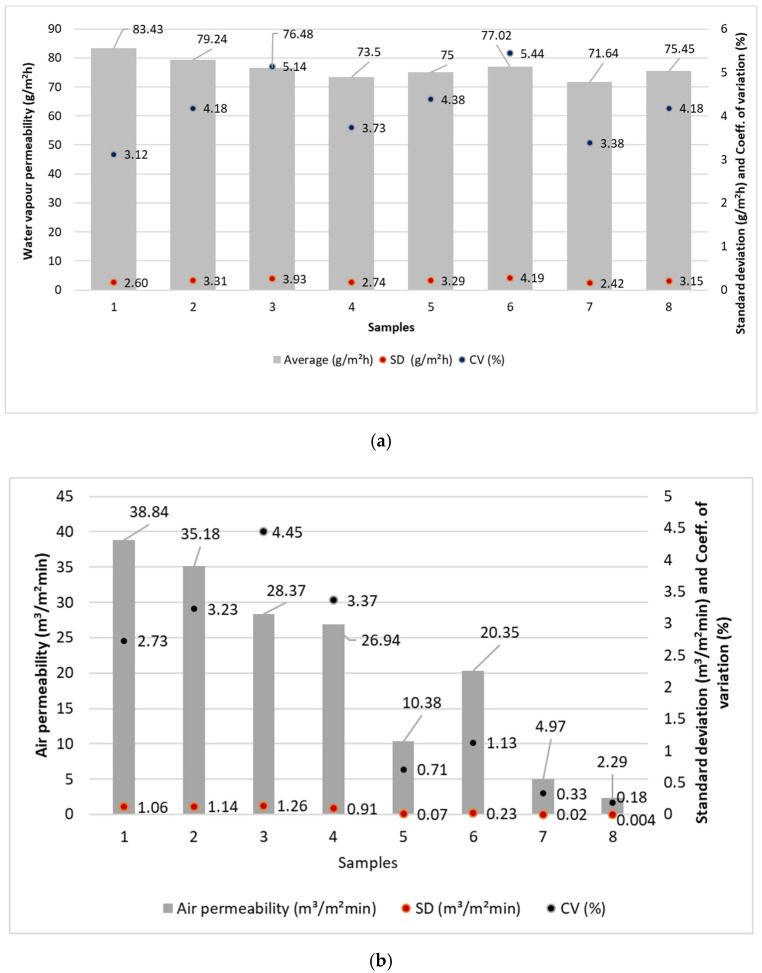
Water vapour permeability (**a**) and air permeability (**b**) of the samples analysed; WVT—water vapour permeability of the samples, g/m^2^h; Q—air permeability, m^3^/m^2^min; SD—standard deviation in g/m^2^h and m^3^/m^2^min; CV—coefficient of variation, %.

**Figure 5 polymers-14-02258-f005:**
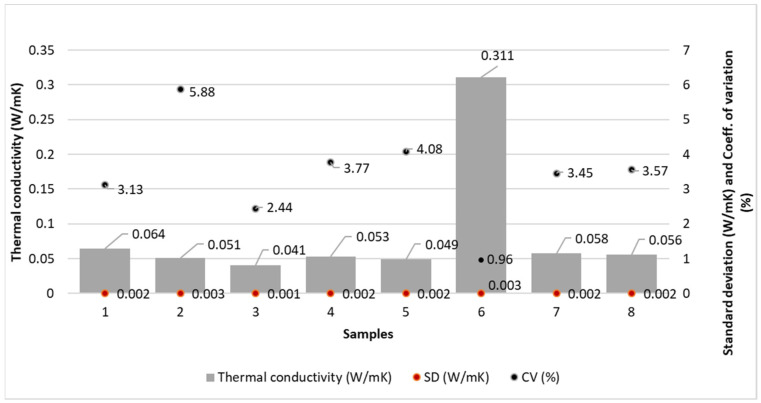
Thermal conductivity of the samples; λ—thermal conductivity, W/mK; SD—standard deviation in W/mK; CV—coefficient of variation, %.

**Table 1 polymers-14-02258-t001:** Raw material composition and base layer production process.

Sample	Raw Material Composition	Type of Base Layer
1	PP/PE	Single layer spunbond process—S
2	PP/PE	Two-layer spunbond process—SS
3	PP	Three-layer spunbond process—SSS
4	PP	Three-layer spunbond process—SSS
5	PP	Five-layer composite of spunbond and meltblown processes—SSMMS
6	PP	Five-layer composite of spunbond and meltblown processes—SSMMS
7	PE	Single layer meltblown process—MB
8	PU	Single layer meltblown process—MB

SS; SSS—two or three-layer composite, where each layer is made by spunbond process, SSMMS—five-layer composite, where three layers are made by spunbond process and two by meltblown process, S—one layer by spunbond process, MB—one layer by meltblown process.

**Table 2 polymers-14-02258-t002:** Fibre diameter.

Sample		Fibre Diameter	
	Average Value [μm]	Standard Deviation (SD) [μm]	Coefficient of Variation (CV) [%]
**Spunbond process**			
1	19.34	1.021	5.28
2	19.57	0.594	3.03
3	11.53	0.427	3.70
4	17.46	0.602	3.45
**SSMMS process**			
5			
A	2.40	0.08	3.33
B	17.41	0.69	3.99
6			
A	1.86	0.022	1.18
B	12.84	0.462	3.59
**Meltblown process**			
7	2.81	0.074	2.64
8	3.55	0.12	3.59

SSMMS—combination of triple spunbond and double meltblown; A—fibre diameter by spunbond (thinner fibres); B—fibre diameter by meltblown (thicker fibres).

**Table 3 polymers-14-02258-t003:** Thickness and Mass.

Sample	Thickness			Mass		
	Average [Mm]	Standard Deviation; SD [Mm]	Coefficient of Variation; CV [%]	Average Mass [G]	Mass per Unit Area [G/M^2^]	Coefficient of Variation; CV [%]
**Spunbond process**						
1	0.366	0.013	3.50	1.70	30.91	0.61
2	0.236	0.016	6.77	1.45	26.36	1.15
3	0.116	0.008	6.90	0.73	13.27	0.73
4	0.216	0.014	6.48	1.42	25.82	0.49
**SSMMS process**						
5	0.142	0.010	7.04	0.84	15.27	1.23
6	0.063	0.0007	1.10	0.42	7.64	0.77
**Meltblown process**						
7	0.186	0.006	3.22	1.53	27.82	0.43
8	0.172	0.006	3.49	2.57	46.73	0.52

SSMMS—A combination of a triple spunbond and a double meltblown process.

**Table 4 polymers-14-02258-t004:** Microscopic appearance of the samples.

Sample	Display of Samples at 25× Magnification	Display of Samples at 2000× Magnification
1	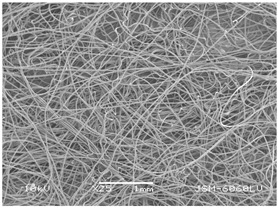	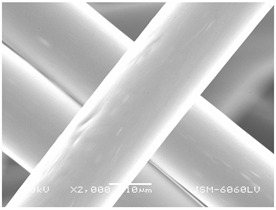
2	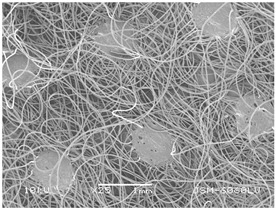	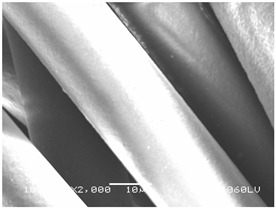
3	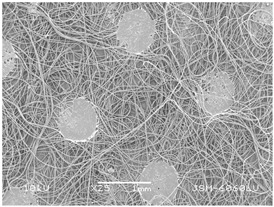	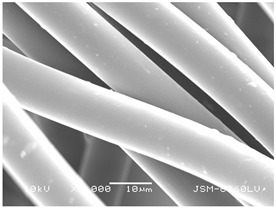
4	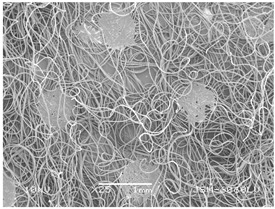	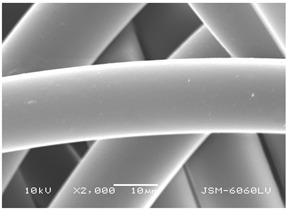
5	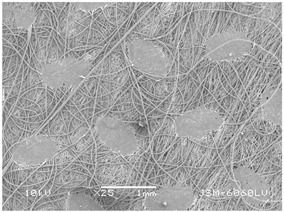	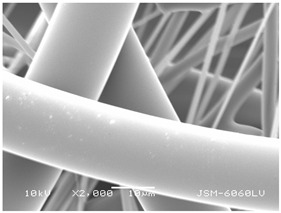
6	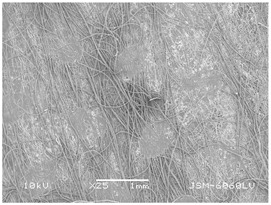	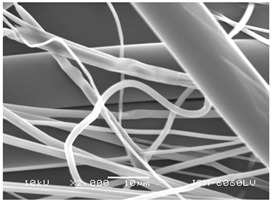
7	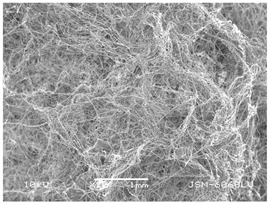	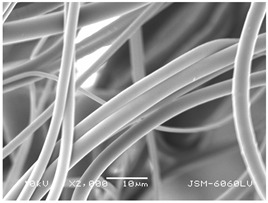
8	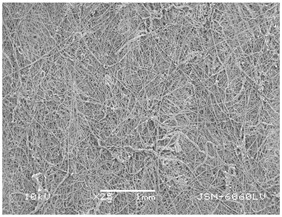	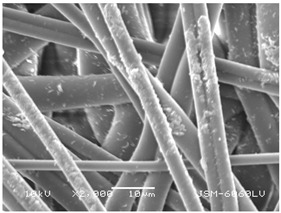

**Table 5 polymers-14-02258-t005:** Porosity of the samples.

Porosity Parameters	Samples							
	1	2	3	4	5	6	7	8
Average pore diameter (µm)	114.84	66.85	73.78	184.70	135.31	133.91	198.59	23.99
Openness of surface (%)	20.95	7.92	30.97	45.19	47.68	29.85	70.77	3.17
Bubble point (µm)	816	667	397	612	408	358	1049	90

**Table 6 polymers-14-02258-t006:** Results of single factor ANOVA.

ANOVA for Breaking Stress Results						
*Source of Variation*	*SS*	*df*	*MS*	*F*	*p-value*	*F-crit*
Between Groups (technology of web formation process)	15.43	3	7.77	3.60	0.045	3.467
**ANOVA for Breaking Extension Results**						
*Source of Variation*	*SS*	*df*	*MS*	*F*	*p-value*	*F-crit*
Between Groups (technology of web formation process)	25147.9	2	12573.9	3.91	0.036	3.467
**ANOVA for Water Vapour Permeability Results**						
*Source of Variation*	*SS*	*df*	*MS*	*F*	*p-value*	*F-crit*
Between Groups (technology of web formation process)	42.7	2	21.3	2.67	0.123	4.257
**ANOVA for Air Permeability Results**						
*Source of Variation*	*SS*	*df*	*MS*	*F*	*p-value*	*F-crit*
Between Groups (technology of web formation process)	1665.9	2	832.9	37.145	4.48⋅10^−5^	4.256
**ANOVA for Thermal Conductivity Results**						
*Source of Variation*	*SS*	*df*	*MS*	*F*	*p-value*	*F-crit*
Between Groups (technology of web formation process)	0.041	2	0.02	1	0.12	4.256
**ANOVA for Porosity-Openness of Surface Results**						
*Source of Variation*	*SS*	*df*	*MS*	*F*	*p-value*	*F-crit*
Between Groups (technology of web formation process)	365.8	2	182.9	0.29	0.75	4.25

*Sum-of-squares (SS) column with no repeated measures, Degrees of freedom (df), Mean squares (MS), F-ratio (F), p-value, F-critical (F-crit).*

## Data Availability

The data presented in this study are available on request from the corresponding author.

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
