# Peer review of "Influence of the Web Formation of a Basic Layer of Medical Textiles on Their Functionality"

_polymers, 2022, doi:10.3390/polym14112258_

Round 1
Reviewer 1 Report
It would be important to provide a detailed review of the main characteristics that medical textiles should have, in order to clarify the combinations that emerge from the tests carried out.
Then, reference should be made to the polluting power of synthetic materials used as microplastics in waste.
Author Response
Answers to the Reviewer 1
Thank you very much for your review.
I have partially incorporated your suggestions in the Abstract and Theoretical part, lines 153 to 163. I have also added to subchapter 2.2. Medical masks and multilayer textiles, lines 198 to 137.
Other changes in the article are made based on the suggestions of reviewers.
All changes are marked in the article with the "Track changes" feature.
Kind regards,
Authors
Reviewer 2 Report
It is a very good study with overall adequate presentation of experimental results. Some additions are needed:
1) Authors should further emphasize on the novelty of their work.
2) Some minor typos, grammar and syntax errors should be carefully revised and corrected accordingly.
3) Reference can be even more updated (more recent relative works).
4) Standard deviation (error bars) should be added in all Figures
Author Response
Answers to the Reviewer 2
Thank you very much for your comments and review.
- Authors should further emphasize on the novelty of their work.
The new insights were included in the Conclusion, line 769 to 777
- Some minor typos, grammar and syntax errors should be carefully revised and corrected accordingly.
The paper was re-reviewed by a lecturer working at our faculty.
- Reference can be even more updated (more recent relative works).
We have supplemented the article with newer and relevant references.
4) Standard deviation (error bars) should be added in all Figures
The standard deviation was included in all graphical representations in the article (except in Figure 3 no, as the figure shows the stress-extension curves).
All changes in the article were inserted with the »Track changes« function.
Kind regards,
Authors

Reviewer 3 Report
This paper describes the formation of the base web for medical textiles and the effect on its function. This is a topic of interest to researchers in related fields, but the paper requires significant improvements to be accepted for publication. My detailed comments are as follows:
1, All image quality of the manuscript needs to be improved. For example, color contrast and sharpness, etc.
2, In section "3.3. Display of used materials on SEM microscope", the ruler of the SEM image should be clearer.
3, In section "3.4. Breaking stress and extension", the author can add stress-strain curves.
4, The conclusion part of the manuscript needs to be refined.
5, The authors are suggested to combine some paragraphs in the article to make it more concise.
6, The format of all references should be consistent.
7, The quality of English needs to be further polished.
Author Response
Answers to the Reviewer 3
Thank you very much for your review and comments.
Comments and answers:
- All image quality of the manuscript needs to be improved. For example, color contrast and sharpness, etc.
Based on your comment, all the images in the article have been improved.
2, In section "3.3. Display of used materials on SEM microscope", the ruler of the SEM image should be clearer.
In Section 3.3, we inserted the original SEM images to make the scale more clearly visible on the images.
3, In section "3.4. Breaking stress and extension", the author can add stress-strain curves.
In Section 3.4, we added stress-extension curves in the longitudinal and transverse directions for the analyzed samples.
4, The conclusion part of the manuscript needs to be refined.
We supplemented the Conclusion in the article and changed it to a more appropriate form.
5, The authors are suggested to combine some paragraphs in the article to make it more concise.
We have combined sections 2.1 and 2.2 to present the requirements and materials for medical masks in a common subsection.
6, The format of all references should be consistent.
We supplemented and corrected the references in accordance with the instructions of the journal.
7, The quality of English needs to be further polished.
The article was reviewed again by a lecturer in our department.
All changes in the article were inserted with the »Track changes« function.
Kind regards,
Authors

Round 2
Reviewer 2 Report
All my comments of the initial submission have been correctly replied and included in the revised manuscript. The quality of this work has been drastically improved after revision and therefore I recommend its publication as it is.